# Eating Disorder Attitudes and Body Self-Image of Urban Mediterranean Adolescents

**DOI:** 10.3390/children10060962

**Published:** 2023-05-29

**Authors:** Francisca Sempere-Ferre, Dolores Escrivá, Jordi Caplliure-Llopis, Inmaculada Benet, Carlos Barrios

**Affiliations:** 1Human Nutrition and Dietetics Department, School of Medicine and Health Sciences, Catholic University of Valencia San Vicente Mártir, 46001 Valencia, Spain; 2Nursing Department, School of Medicine and Health Sciences, Catholic University, 46007 Valencia, Spain; dolores.escriva@ucv.es (D.E.); jorcallo@mail.ucv.es (J.C.-L.); 3Institute for Research on Musculoskeletal Disorders, School of Medicine and Health Sciences, Catholic University of Valencia San Vicente Mártir, 46001 Valencia, Spain; carlos.barrios@ucv.es; 4Surgical Nursing Division, Valencia Clinic Hospital, 46010 Valencia, Spain; inmaculada.benet@mail.ucv.es

**Keywords:** eating disorders, anthropometric measures, body dissatisfaction, EDI-3, adolescents, statistics

## Abstract

Background: Early detection tools for eating disorders play an essential role in their prevention. The aim of this study was to analyze different variables associated with the risk of eating disorders and their relation to anthropometric parameters, nutritional status and body self-image. Methods: The Eating Disorder Invetory-3 Referral Form (EDI-3 RF) and the Body Shape Questionnaire (BSQ) were administered to 809 adolescents (413 boys and 396 girls), aged 11 to 17 years, from three randomly chosen schools in a big Mediterranean city. Anthropometric measurements were taken following standardized methods. Overweight and obesity were determined according to the International Obesity Task Force (IOTF). Results: Overweight (23%) prevailed over obesity (9.37%). Girls registered a higher score in the Obsession for Thinness EDI-3 RF subscale and in the body dissatisfaction section of the BSQ. The only statistically significant differences in the Obsession for Thinness and Body Dissatisfaction levels among the different ages were seen in boys. In this series, body dissatisfaction assessed by the EDI-3 RF was not correlated with BSQ body dissatisfaction. Sex and age in adolescence determine the relationship between eating disorder attitudes and body shape dissatisfaction. Conclusions: It is necessary to look for other parameters to investigate to determine body dissatisfaction.

## 1. Introduction

Eating disorders (EDs) are complex psychiatric illnesses characterized by the excessive preoccupation with ingested food, use of unhealthy measures to control or reduce body weight, and dissatisfaction with the perception of body image [1]. The last classification made by the American Psychiatric Association in the Diagnostic and Statistical Manual of Mental Disorders (DSM-5) in 2013 established different forms of EDs: three typical eating disorders, anorexia nervosa, bulimia nervosa, and binge eating disorder; other specific feeding or eating disorders (OSFED); and unspecified feeding or eating disorders (UFED) [2,3].

The etiology of this group of diseases is complex, and different studies have observed that multiple sociocultural, family, genetic, personal, and other factors seem to influence its development. Epidemiological studies carried out to date establish that members of the female sex are more vulnerable to these disorders, and that they are more frequent in adolescence [4].

In the 21st century society, EDs constitute a public health problem, not only because of the number of people affected, which increases annually, but also because of the duration of the treatments and the associated psychological and physical comorbidities. These comorbidities can become chronic, and can cause death [5]. In addition, there are several prevalence studies indicating that eating disorders are underdiagnosed, and that there are many more people who suffer from them or who, due to certain factors, could acquire them in the future [6,7].

It is for this reason that the use of early detection tools play a fundamental role. There are different methods, such as structured interviews; self-reporting questionnaires like the Eating Attitudes Test 26 (EAT-26), the Eating Disorder Examination Questionnaire (EDE-Q), the SCOFF questionnaire, the Short Evaluation of Eating Disorders (SEED), and the Eating Disorder Inventory (EDI-3); and body perception tools like the Body Shape Questionnaire (BSQ), the Body Attitude Test (BAT), and the Body Appreciation Scale (BAS); however, they can all minimize the impact of these diseases on society [8,9,10,11,12].

The evaluation of body dissatisfaction is one of the most common aspects in the study of eating disorders, since it is a characteristic that defines people who suffer from this group of diseases. Body dissatisfaction corresponds to the negative perception of the image that one has about their body, and is usually common during adolescence where changes occur at physical, psychological, and cognitive levels. [13,14]. The self-administered Body Shape Questionnaire is one of the most used tools for its evaluation worldwide [15,16,17,18]. In addition, the somatotype that quantifies the current morphological conformation of the human body could be a good predictor of these diseases. The endomorph component indicates a tendency towards obesity, the mesomorph component is dominated by skeletal muscle mass, and the ectomorph component corresponds to the long-limbed and asthenic types [19].

Little is known as to how disordered eating attitudes are related to different associated variables such as body mass index and other anthropometric measures, or the relationship between different tools used for the evaluation of the same parameters in at-risk populations. However, there are many studies that have investigated predisposing factors to these disturbances in adolescents, such as psychological aspects [20].

The current study aimed to (1) analyze the different anthropometric measures and somatotypes of Mediterranean adolescents; (2) evaluate the scores obtained by adolescents on the three risk subscales of the EDI-3 Referral Form; (3) study the level of body dissatisfaction of the participants through the Body Shape Questionnaire; (4) establish the possible relationships between anthropometric measures, somatotypes, and the four categories established by the International Obesity Task Force with the Obsession for Thinness, Bulimia and Body Dissatisfaction risk subscales; (5) Investigate the association between the scores obtained on the Body dissatisfaction scale from the EDI-3 Referral Form and those obtained from the Body Shape Questionnaire.

## 2. Materials and Methods

### 2.1. Study Design

A cross-sectional study was conducted in three schools in the city of Valencia (Spain). The centers were randomly selected among those with an agreement with the Catholic University of Valencia San Vicente Mártir for the internship of their students in the Education program. Parents signed an informed consent form after receiving a detailed explanation of the project. All adolescents also expressed their agreement to participate in the study. The study was carried out between September and June 2018 and 2019 and data were collected in the same centers.

### 2.2. Participants

A random sample of 809 students were recruited for the study. Inclusion criteria included healthy adolescents aged between 11 and 17 years and who played sport more than 6 h per week. Students with chronic diseases that could affect the values of anthropometric measurements were excluded.

### 2.3. Data Measurements

#### 2.3.1. Anthropometric Data

Anthropometric measurements were recorded in the different schools. All participants were measured without shoes and minimal clothing to the nearest 0.1 cm (SECA 225, SECA, Hamburg, Germany), and weighed to the nearest 0.1 kg (SECA 861, SECA, Hamburg, Germany).

The body mass index (BMI) was calculated as weight divided by height squared. Seven skinfold thicknesses (biceps, triceps, subscapular, abdominal, suprailiac, thigh and calf) were measured in triplicate on the right side of the body with a skinfold caliper (Holtain Ltd., Gales, UK) following the recommendations of the International Society for the Advance of Kinanthropometry (ISAK) [21]. A sole well-trained investigator (level 2 ISAK certified) recorded all the measurements to minimize methodological bias. The mean of three measures was taken as the valid measure.

Body fat mass and percentage of body fat (BF%) were obtained from measurements of two skinfold equations for the general population, and a specific body-fat predicting equation for soccer players [22,23]. The somatotype components (endomorphy, mesomorphy, and ectomorphy) of each participant were assessed using the technique of Carter [24].

Based on the BMI, and taking into account the cut-off points proposed by the International Obesity Task Force (IOTF), the participants were classified into four categories: underweight, normal weight, overweight and obese [25].

#### 2.3.2. Analysis of Eating Disorder Risk Subscales

The EDI-3 Referral Form Questionnaire for the Spanish population was used in this study. Participants were asked to answer 25 questions about the three risk subscales: Obsession for Thinness (EDI-OT), Bulimia (EDI-B) and Body Dissatisfaction (EDI-BD) (Cronbach’s alpha between 0.74 and 0.96) [26].

Each question consisted of a 6-point Likert scale (never, rarely, sometimes, often, almost always, always). Each item was scored from 0 to 4. Extreme responses in the pathological or symptomatic direction of the item (always or never, depending on the direction of the item) received a score of 4. The response options immediately adjacent to the above (almost always or rarely) received a score of 3, while the following ones (sometimes and often) received 2 points and the next two adjacent options (often and sometimes) received a score of 1. The two options for responses opposite to the pathological pole of the item both received a score of 0.

To calculate the total direct scores for each of the Obsession for Thinness, Bulimia and Body Dissatisfaction subscales, the values of each of the items that made them up were added. For the evaluation of the EDI BD risk subscale, three qualitative levels of dissatisfaction with one’s own body were established: lower (0–6), moderate (7–27), and high (28 or more) [26].

#### 2.3.3. Body Self-Image Evaluation

The Body Shape Questionnaire, with transcultural adaption for Spanish adolescents, was the instrument selected for body image assessment [27,28]. The reliability of this questionnaire for the young Spanish population has been confirmed, with high internal consistency obtained in a study of the adaptation (Cronbach’s alpha between 0.74 and 0.96) [27,28].

Each of the 34 questions of the BSQ had a Likert-like self-reporting scale, with six response options (1 = never, 2 = rarely, 3 = sometimes, 4 = often, 5 = very often, 6 = always). Body dissatisfaction was tested through 12 questions; fear of gaining weight, 10 questions; feelings of worthlessness because of appearance, 10 questions; and desire to lose weight, 6 questions. Four levels of dissatisfaction with physical appearance were considered and was scored as follows: no concern about your image(below 40); concern about your weight (40–50); body dissatisfaction (51–101); body image disorder (102–204) [28].

### 2.4. Data Analysis

Anthropometric data and scores from the questionnaires were evaluated using the statistical software SPSS v.20 (IBM Corp., Armonk, NY, USA). A descriptive analysis was performed using frequency, mean values, and standard deviations. The normality of the variable distribution was verified by the Kolmogorov–Smirnov test. The Chi-square test was applied to examine the associations of sex and age with the different categories established by the Body Shape Questionnaire. A regression test (Pearson’s r) and Spearman test were used to establish correlations between the different EDI-3 RF subscales; between these, the different anthropometric variables, the IOTF groups and between the BSQ and IOTF groups, and the risk subscales of the EDI-3 Referral Form. Internal consistency or the scales were explored using the Cronbach’s Alpha coefficient. The significance level used for the different analyses was (α = 0.05).

## 3. Results

### 3.1. Sample Characteristics and Evaluation of Body Measures, Somatotypes, Overweight and Obesity

Of the sample of 809 students, 413 were boys (51.1%) and 396 girls (48.9%), with a mean age of 13.7 years. Two hundred and seventy-six (34.1%) of the participants were between 11 and 12 years old, 346 (42.7%) between 13 and 15 years old, and 187 (23.1%) between 16 and 17 years old.

Table 1 shows the data related to height, weight, body mass index (BMI), body surface, fat mass index (FMI), muscle and fat mass, and their percentages, corresponding to the whole sample distributed by sex. The table also lists the results obtained for the different somatotypes. Somatotype analysis between boys and girls revealed slightly higher values of endomorphic and ectomorphic patterns in girls and mesomorphic in boys (Table 1). Table 2 displays the height, weight, and BMI according to age and sex including the 25th, 50th, and 75th percentiles.

The analysis of the incidence of overweight and obesity by sex and age through the IOTF Guidelines showed that overweight prevailed over obesity in all the groups analyzed. Boys registered percentages of overweight and obesity of 11.38% and 5.08%, respectively, while in girls these were 11.62% and 4.29%. In the 11-to12-year age range, a higher percentage of boys were overweight (13.51%), while obesity was very similar between both sexes. Between 13 and 15 years, overweight prevailed in girls (11.48% versus 8.59%), while obesity was also very similar for both groups. At 16–17 years, the incidences of overweight and obesity were registered at 12.62% and 5.83% in boys and 13.10% and 3.57% in girls.

### 3.2. Obsession for Thinness, Bulimia and Body Dissatisfaction

The analysis of the results obtained in the EDI-3-RF items that evaluated body dissatisfaction showed that it was moderate in most adolescents (73.8% in boys and 87.6% in girls). It was only high in 9.9% of the boys and in 2.5% of the girls. Girls registered a higher score when the Obsession for Thinness risk subscale was evaluated, at 11.6 compared to 8.7 in boys. Statistical analyses showed that between both sexes’ scores, there were statistically significant differences (*p* < 0.05). Values recorded for the risk factors EDI-B (Bulimia) and EDI-BD (Body Dissatisfaction) did not differ practically between boys and girls (Table 3). Cronbach’s alpha for measuring the three subscales of eating disorders were acceptable (EDI-OT = 0.77, EDI-B = 0.72, and EDI-BT = 0.78).

Table 4 lists the Obsession for Thinness, Bulimia and Body Dissatisfaction scores at different ages. These varied between the female and male sex. For example, the highest score in EDI-BD in boys was registered at 11 years old, while for girls it was at 15. At this age, they registered the highest scores on all subscales.

All the correlations between the different risk subscales were significant (*p* < 0.05) (Table 5).

### 3.3. Relation between the EDI-3 RF Risk Subscales, Anthropometric Measures, Somatotypes and IOTF Groups

When the analysis regarding a relationship between the anthropometric variables and Obsession for Thinness, Bulimia and Body Dissatisfaction subscales was conducted, it was observed that there was a slight significant correlation between EDI-OT and percentage of body fat, lean weight and EDI-BD, and the different somatotypes with EDI-BD (Table 5). No significant correlation was found when the relationship between EDI-BD, EDI-B, EDI-OT and the different groups established by the IOTF was studied. Analyzing the whole sample, the three EDI subscales were inversely correlated with age. However, this correlation applied only for boys (EDI-BD: r = −0.236; *p* < 0.001; EDI-B: r = −0.107; *p* > 0.05; EDI-OT: r = −0.141; *p* < 0.01) and no for girls (EDI-BD: r = −0.087; *p* = 0.117; EDI-B: r = −0.059; *p* = 0.288; EDI-OT: r = −0.022; *p* = 0.697).

### 3.4. Body Shape Questionnaire

The analysis of the values in the Body Shape Questionnaire showed that 267 of the adolescents did not show concern for their image (35.59 ± 1.79), 120 expressed concerns about their weight (44.85 ± 3.14), 350 registered body dissatisfaction (69.15 ± 14.89), and 72 demonstrated body image disorders (120.52 ± 19.38). Cronbach’s alpha for the Body Shape Questionnaire were acceptable (0.75).

By sex, 3.4% of the boys did not show concern for their image compared to 26.2% of the girls. The percentages that expressed concern about their weight were very similar in both sexes. According to the scores obtained from the questionnaire, 10.4% of the girls and 7.6% of the boys registered a body-image disorder. Most of the girls expressed body dissatisfaction (47.8%). There was a significant association between the Body Shape Questionnaire groups and the sex variable (χ^2^ = 17.983, *p* = 0.0004).

Between 11 and 13 years of age, most of the participants did not show concern about their image, while in the age range between 14 and 17, the most prevalent option was body dissatisfaction. there was a significant relationship between the different items of the questionnaire and the ages (χ^2^ = 39.574, *p* = 0.0024). No significant correlations between BSQ scores and IOTF groups were registered.

### 3.5. EDI-3 Body Dissatisfaction and BSQ Questionnaire Relationship

The Pearson coefficient correlation did not show any relationship between the scores obtained on the EDI-BD risk subscale and those obtained on the BSQ questionnaire for either sex (Table 5).

## 4. Discussion

The current study analyzed the anthropometry profile of a sample of healthy Mediterranean adolescents between 11 and 17 years old and their relationships with their nutritional status, body dissatisfaction, and the different risk subscales of the Eating Disorder Inventory-3 RF. Several investigations have been carried out in Spain addressing the prevalence of eating disorders in adolescents using different tools, but none has investigated its relationship with nutrition status and body dissatisfaction in this population group [28,29,30]. The results showed that these relationships were conditioned by age and sex.

Adolescence is a period of transition where a series of mental and physical changes take place, being a critical time for the development of eating disorders [31]. The prevalence of these disturbances is usually analyzed through self-assessed questionnaires, administered to the adolescents themselves, assessing behaviors that may in some cases evolve into an eating disorder or be related to psychological distress, conditioning intake, somatic development and their quality of life [32]. In Spain, studies on the risk of suffering from eating disorders in adolescents have recorded values of 9.2%, 14.32%, 11.2% and 27% in different geographical areas such as the Community of Madrid, Tarragona, Alicante and the Canary Islands, respectively [33,34,35,36].

Body dissatisfaction is one of the key predisposing factors for eating disorders [20,37,38]. In this research, female participants showed greater dissatisfaction with their bodies than males, although the percentages recorded in the two tools used (EDI 3-RF and BSQ) were different. Studies show that this dissatisfaction is more common among adolescent girls because, among other factors, at the onset of puberty there is an increase in weight resulting in a higher proportion of body fat [39,40,41]. It could also be attributed to the fact that pubertal changes tend to be later in boys [42].

The social pressure exerted by the different environments within which adolescents move, including family, school, peers, media, social networks, fashion and cinema, makes adolescents more susceptible to an idealized and unrealistic body stereotype [40,43]. To achieve this ideal appearance, adolescents try to control their weight, often resulting in inappropriate behaviors such as restrictive dieting, fasting, self-inflicted vomiting after uncontrollable overeating or the use of laxatives and diuretics. The subscales of the EDI-OT and EDI-BD measure some of these tendencies. In the current study, the results obtained using these constructs only recorded a statistically significant difference between the mean values on Obsession for Thinness when comparing the values obtained for both sexes. This divergence could be due to the social influence with ideal body types, as girls lean towards slimmer body shapes and therefore engage in what they perceive as weight-reducing behaviors, while boys prefer to have a muscular body and wish to gain weight. In addition, teenage boys tend to underestimate their weight while girls tend to overestimate their weight [39].

Concerning anthropometric measures, elevated BMI and body fat mass percentage have been reported to be associated with an increased risk of eating disorders. In this sense, the association between the values obtained from the EDI 3-RF and BSQ and other questionnaires, with the surveyed anthropometric measurements, has been investigated. Thus, a great body dissatisfaction has been associated with high BMI [37,43,44]. In the current study, the body dissatisfaction EDI-3 RF subscale (EDI-BD) correlated with the different somatotypes and lean mass. The Obsession for Thinness Subscale (EDI-OT) correlates with % of body fat and the Fat Mass Index. Perhaps the values obtained from the different questionnaires could be higher if the inclusion criterion had not been that children exercised at least 6 h per week.

Different investigations have evaluated the relationship of BSQ with other tools that measure body dissatisfaction, or some aspects related to it. In this study, there was no relationship between the two tools used to study body dissatisfaction (Body Shape Questionnaire and EDI-BD). This variation in the different measures of body dissatisfaction could be explained by the fact that the BSQ is an instrument that mainly covers emotional, cognitive, perceptual and behavioral aspects of body dissatisfaction, whereas the EDI sub-item assesses more affective and cognitive nuances.

These results obtained in this Mediterranean population of healthy adolescents did not agree with those found by other researchers when they compared both tools, as they demonstrated that there was already a high correlation between the BSQ and the subscales of EDI-1 and EDI-2 [45,46,47]. In a study by Campana et al. who examined 20 people with eating disorders, it was observed that the results obtained in the BSQ, the Body Checking Questionnaire (BCQ), and the Body Image Avoidance Questionnaire (BIAQ) had a high correlation [48].

There were several limitations in the current study. At a methodological level, the transversal nature of the work did not allow us to make causal inferences that could interact in the results obtained. Another limitation is that changes in body composition in growing children and adolescents must be obtained in longitudinal studies, that give the possibility of evaluating natural variations in individual growth and development. Some studies have shown that when self-administered questionnaires are used in adolescents, they are not usually answered reliably [49]. However, Fortes et al. noted that these tools can be very effective in studies with large samples, as they are easy to use and inexpensive [50]. Finally, the presence of psychiatric symptoms such as depression, anxiety, etc., and other factors, which may affect some variables considered in this study, such as body dissatisfaction, were not assessed [51,52,53].

## 5. Conclusions

The results of the present study suggest that the relationship between the different tools for the early detection of eating disorders is conditioned by population group, interacting with different factors such as age and sex. It is necessary to look for other parameters to investigate their impact on body dissatisfaction. In the current study, it has been observed that two scales measuring the same variable—body dissatisfaction—contemplate different aspects.

In summary, a message for clinicians working with these adolescents is that body image dissatisfaction is not a good indicator of eating disorders. Body image dissatisfaction was not related to nutritional status, particularly in girls. Some other psychosocial factors beyond anthropometry merit further research to explain the lack of correlation between body image concerns and nutritional status specifically in girls.

## Figures and Tables

**Table 1 children-10-00962-t001:** Anthropometric measures and somatotypes of Mediterranean adolescents included in the whole sample and distributed by sex. SD: Standard deviation, CI: Confidence interval, BMI: body mass index, FMI: fat mass index.

	Boys n = 413		Girls n = 396	
	Mean ± SD	95% CI	Mean ± SD	95% CI
**Age (years)**	13.7 ± 2.0	13.57–13.96	13.7 ± 1.9	13.55–13.92
**Height (cm)**	157.8 ± 10.9	156.7–158.8	155.9 ± 8.3	155.1–156.8
**Body mass (kg)**	50.8 ± 13.9	49.49–52.18	48.41 ± 10.7	47.34–49.46
**BMI (kg/m^2^)**	20.1 ± 3.5	19.76–20.43	19.69 ± 3.0	19.4.5–19.99
**Body surface**	1.65 ± 0.3	1.4–1.7	1.61 ± 0.2	1.3–1.8
**Fat mass (kg)**	7.6 ± 4.7	7.10–8.01	7.6 ± 3.6	7.21–7.94
**%Body fat**	14.1 ± 4.7	13.66–14.58	15.1 ± 4.1	14.65–15.44
**FMI**	2.9 ± 1.6	2.81–3.12	3.1 ± 1.3	2.93–3.19
**Muscle mass (kg)**	23.6 ± 5.7	23.04–24.15	24.1 ± 4.8	23.61–24.57
**% Body muscle**	46.9 ± 3.7	46.43–47.16	50.1 ± 3.2	49.73–50.38
**Endomorphy**	3.5 ± 1.8	3.32–3.67	4 ± 1.5	3.85–4.16
**Mesomorphy**	2.7 ± 1	2.62–2.83	1.9 ± 1	1.86–2.07
**Ectomorphy**	3.1 ± 1.5	2.86–3.15	3.0 ± 1.3	2.92–3.18

**Table 2 children-10-00962-t002:** Height, weight, and BMI according to age and sex. SD: Standard deviation.

	Boys		Percentile			Girls		Percentile		
	n	Mean ± SD	25th	50th	75th	n	Mean ± SD	25th	50th	75th
Height/Age										
**11 yr**	62	147.2 ± 5.3	143.0	146.5	151.2	52	145.4 ± 4.3	143.0	145.0	148.0
**12 yr**	85	148.9 ± 7.2	143.0	148.0	155.0	77	149.8 ± 6.4	145.0	151.0	154.0
**13 yr**	64	154.7 ± 6.7	151.0	153.0	159.0	64	156.8 ± 6.9	153.0	156.0	160.7
**14 yr**	36	161.0 ± 9.4	155.2	162.0	168.7	57	159.5 ± 6.4	155.5	160.0	164.0
**15 yr**	63	164.4 ± 7.5	160.0	165.0	169.0	62	159.0 ± 5.2	156.0	158.0	161.0
**16 yr**	48	168.2 ± 7.7	162.2	168.5	173.0	44	163.6 ± 6.4	160.0	162.0	168.0
**17 yr**	55	168.4 ± 7.2	162.0	168.0	171.0	40	162.2 ± 2.9	161.0	163.0	164.0
Weight/Age										
**11 yr**	62	39.8 ± 6.5	34.4	37.3	46.5	52	37.7 ± 5.1	34.6	35.9	41.1
**12 yr**	85	42.8 ± 9.2	35.3	42.9	49.3	77	40.8 ± 6.8	36.6	40.8	45.1
**13 yr**	64	45.6 ± 9.3	38.3	44.5	51.2	64	50.8 ± 12.5	41.8	46.5	58.7
**14 yr**	36	54.2 ± 11.6	46.2	54.5	62.8	57	51.4 ± 9.9	44.0	50.0	58.2
**15 yr**	63	56.6 ± 9.9	50.2	54.7	59.9	62	52.7 ± 9.5	46.9	51.2	54.4
**16 yr**	48	63.2 ± 16.3	53.5	59.8	69.1	44	55.8 ± 8.6	51.4	54.1	61.0
**17 yr**	55	62.1 ± 13.1	53.8	56.8	65.1	40	53.8 ± 3.8	51.7	54.1	55.8
BMI/Age										
**11 yr**	62	18.4 ± 2.8	16.4	17.8	20.5	52	17.8 ± 2.1	16.1	17.4	19.2
**12 yr**	85	19.1 ± 2.9	17.2	19.1	21.2	77	18.1 ± 2.3	16.8	17.7	19.3
**13 yr**	64	18.9 ± 2.7	16.7	18.4	20.7	64	20.4 ± 3.5	17.6	19.8	22.5
**14 yr**	36	20.7 ± 3.1	18.3	19.9	23.7	57	20.2 ± 3.6	17.8	19.9	22.3
**15 yr**	63	20.9 ± 2.8	19.0	20.5	22.1	62	20.8 ± 3.0	18.3	20.3	22.0
**16 yr**	48	22.1 ± 4.1	19.3	20.9	24.7	44	20.8 ± 2.3	19.3	20.3	22.0
**17 yr**	55	21.8 ± 4.1	19.3	20.9	22.1	40	20.5 ± 1.5	19.9	20.7	21.8

**Table 3 children-10-00962-t003:** Means and standard deviations of the Obsession for Thinness (EDI-OT), Bulimia (EDI-B) and Body Dissatisfaction (EDI-BD) risk subscales from the EDI-3-Referral Form by sex.

	EDI-OT	EDI-B	EDI-BD
**Boys**	8.7 ± 6.7	5.2 ± 5.3	17.2 ± 8.5
**Girls**	11.6 ± 6.8	5.7 ± 5.2	17.4 ± 6.8
**Total**	9.9 ± 6.9	5.4 ± 5.3	17.3 ± 7.8

**Table 4 children-10-00962-t004:** Means and standard deviations of the risk subscales from the EDI-3-Referral Form by age and sex. EDI-OT: Obsession for Thinness, EDI-B: Bulimia, EDI-BD: Body Dissatisfaction.

	Boys			Girls		
Age	EDI-OT	EDI-B	EDI-BD	EDI-OT	EDI-B	EDI-BD
**11**	10.40 ± 7.00	6.87 ± 6.38	19.30 ± 7.06	11.69 ± 6.06	5.51 ± 5.69	16.98 ± 6.56
**12**	10.58 ± 7.63	4.98 ± 4.97	20.41 ± 5.57	11.30 ± 6.00	6.04 ± 5.58	18.34 ± 5.67
**13**	7.90 ± 6.62	5.68 ± 5.61	16.26 ± 8.73	13.51 ± 7.14	6.04 ± 5.01	17.98 ± 6.37
**14**	6.94 ± 4.22	4.61 ± 5.13	18.77 ± 8.16	10.47 ± 5.88	5.56 ± 5.22	17.00 ± 6.64
**15**	6.52 ± 5.87	4.79 ± 5.36	15.73 ± 9.41	11.05 ± 6.43	6.05 ± 6.27	19.88 ± 5.32
**16**	8.14 ± 6.80	4.41 ± 3.89	13.25 ± 9.45	9.47 ± 7.56	6.66 ± 5.09	14.71 ± 7.25
**17**	8.61 ± 6.38	4.76 ± 5.24	15.21 ± 9.64	12.35 ± 8.65	4.12 ± 3.29	15.90 ± 9.32
*p*	0.0019 *	0.3224	0.0002 *	0.1824	0.5519	0.2337

* *p* < 0.05.

**Table 5 children-10-00962-t005:** Matrix of Pearson’s correlations (r) between the different EDI-3 RF risk subscales, the anthropometrics measures, the groups stablished by the International Obesity Task Force (IOTF), the Body Shape Questionnaire (BSQ) and age. EDI-OT: Obsession for Thinness, EDI-B: Bulimia, EDI-BD: Body Dissatisfaction.

EDI-3 RF Risk Subscales	EDI-BD	EDI-OT	EDI-B
**EDI-BD**		0.373 *	0.205 *
**EDI-OT**	0.373 *		0.200 *
**EDI-B**	0.205 *	0.200 *	
**Anthropometric measures**			
**BMI**	−0.051	0.69	−0.035
**%Body fat**	0.052	0.124 *	−0.013
**Total mass (kg)**	−0.105 **	−0.003	−0.059
**Fat mass (kg)**	−0.017	0.072	−0.023
**FMI**	0.020	0.109 *	−0.017
**Lean mass**	−0.179 *	−0.063	−0.085
**Endomorphy**	0.131 *	−0.024	0.064
**Mesomorphy**	0.136 *	0.065	0.064
**Ectomorphy**	−0.127 *	0.030	−0.003
**BSQ**	0.036	0.994	0.090
**Age**	−0.180 **	−0.107 **	−0.090 *

* Significant correlation at the 0.05 level; ** 0.01 level.

## Data Availability

Raw data are available upon request to corresponding author.

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
