# Peer review of "Eating Disorder Attitudes and Body Self-Image of Urban Mediterranean Adolescents"

_children, 2023, doi:10.3390/children10060962_

Round 1
Reviewer 1 Report
This is a cross-sectional study about eating disorder attitudes and body self-image of urban Mediterranean adolescents.
You did not describe the exclusion criteria from the study. Please add this information.
The percentage is not 100%% (413 were boys (51.05%) and 396 girls (48.94%)). Please correct.
Line 251... correct the parenthesis (35.59 1.79).
Line 295... too much parenthesis.
The references are appropriate, the article presents 44 references. I recommend that you expand the discussions by adding references from the last two years. Reference 18 should be filled in with the page number and the year should be written in bold, just like the other references.
Author Response
Los comentarios a los revisores se agregan en el archivo adjunto.

Reviewer 2 Report
Dear Authors,
Thank you for submitting this paper which is related to eating disorders, body satisfaction, and anthropometrics measurements among Mediterranean adolescents. While the number of participants is good, I have some concerns and comments for this paper below.
Introductıon
- There are short paragraphs in the introduction, which is a repetition of the method section because of the information of the measurements. Since this issue has been studied several times, the authors should give the study's problem with a systematic review and meta-analysis and then add the aim of this study. Please do not remember one or two sentences in not a paragraph is a bullet and merge the paragraphs.
- The third paragraph needs references.
Methods
- The method section should be reorganized with the subtitle aim, study, design, settings (time and place), participants, data measurements, ethical considerations, and data analysis.
- The inclusion and exclusion criteria should be described in the participant section.
- The data measurements need to be clarified. Please add the Cronbach of the scales in the original validity and reliability studies and your current study.
- I could not understand endo/meso/ectomorphology that it needs to clarify in the introduction part.
- The second sentence of the participant part should be removed ethical consideration part.
- How did the authors select these schools? Are they randomized? Please explain.
Results
- Comparing the adolescents' height, weight, and BMI analysis is incorrect. Because the fat and weight of girls and boys are different, they also have different ages to start adolescence. For the analysis, I strongly recommend analyzing according to the percentile of these children. The result section should be reanalyzed and revised according to this statement.
- Results should be rewritten according to APA.
- Instead of comparing the classification of age and scale scores with ANOVA, I recommend analyzing the correlation between age and scales.
Discussion
- The discussion part is superficial and should be rewritten according to the revision of the results and added meta-analysis in this issue. The authors should be emphasized the similarity and differences between this study and the current literature.
Limitations
- The limitations section should be given separately.
Conclusion
- Please add the recommendation for clinicians who work with these adolescents and researchers for future studies.
References
- There are APA and numbered references for the text in the study. The authors should check rules of journal rules for references in the text and at the end.
Author Response

(The authors gave the same response as above.)

Round 2
Reviewer 2 Report
Dear authors,
I appreciate your effort to revise and improve the manuscript. It is much clear now than before. I still strongly recommend using the adolescents' percentiles for the analysis. You stated that you do not have the percentile of the adolescents; the percentile could be calculated with the height and weight of the adolescents.
I also recommend improving the discussion part, it is still superficial.
I wish you success in your work.
Author Response
In response to reviewers' comments, the authors have added a table with percentiles and modified the discussion.
